

# Vitamin D receptor-deficient keratinocytes-derived exosomal miR-4505 promotes the macrophage polarization towards the M1 phenotype

Wen Sun[1,*], Jianqin Chen[2,*], Jingting Li[3], Xiaoguang She[1], Hu Ma[1], Shali Wang[1], Jing Liu[4] and Yuan Yuan[5]

[1] Department of Dermatology, Jingmen Central Hospital, Jingmen, China
[2] Department of Dermatology, Integrated Hospital of Traditional Chinese Medicine, Southern Medical University, Guangzhou, China
[3] Department of Traditional Chinese Medicine, Huazhong University of Science and Technology Union Shenzhen Hospital (Nanshan Hospital), Shenzhen, China
[4] Department of Dermatology, The First Affiliated Hospital of Guangzhou University of Chinese Medicine, Guangzhou, China
[5] Department of Surgical Anesthesiology, Jingmen Central Hospital, Jingmen, China
* These authors contributed equally to this work.

Corresponding authors
Jing Liu, liujing1350@gzucm.edu.cn
Yuan Yuan, MDSunny@yeah.net

## ABSTRACT

**Background:** The vitamin D receptor (VDR) has a low level of expression in the keratinocytes of patients with psoriasis and plays a role in the development of the disease. Furthermore, the crosstalk between macrophages and psoriatic keratinocytes-derived exosomes is critical for psoriasis progression. However, the effects of VDR-deficient keratinocytes-derived exosomes (Exos-shVDR) on macrophages and their underlying mechanisms remain largely unknown.
**Methods:** VDR-deficient keratinocytes were constructed by infecting HaCaT cells with a VDR-targeting lentivirus, mimicking the VDR-deficient state observed in psoriatic keratinocytes. Exosomes were characterized using transmission electron microscopy, nanoparticle tracking analysis, and Western blot. The effect of Exos-shVDR on macrophage proliferation, apoptosis, and M1/M2 polarization was assessed using cell counting kit-8 assay (CCK-8), flow cytometer, real-time quantitative polymerasechain reaction (RT-qPCR), and enzyme-linked immunosorbent assay (ELISA). The mechanism underlying the effect of Exos-shVDR on macrophage function was elucidated through data mining, bioinformatics, RT-qPCR, and rescue experiments.
**Results:** Our results revealed that both Exos-shVDR and Exos-shNC exhibited typical exosome characteristics, including a hemispheroid shape with a concave side and particle size ranging from 50 to 100 nm. The levels of expression of VDR were significantly lower in Exos-shVDR than in Exos-shNC. Functional experiments demonstrated that Exos-shVDR significantly promoted macrophage proliferation and polarization towards the M1 phenotype while inhibiting macrophage apoptosis. Moreover, miR-4505 was highly expressed in the skin tissue of patients with psoriasis. Its overexpression significantly increased macrophage proliferation and polarization towards M1 and inhibited apoptosis. Furthermore, the effects of Exos-shVDR on macrophage function occur through miR-4505.

**Conclusions:** Exos-shVDR exacerbates macrophage proliferation, promotes polarization towards the M1 phenotype, and inhibits macrophage apoptosis by increasing the levels of miR-4505. These results indicate that modulation of macrophage function is a potential strategy for developing new drugs for the treatment of psoriasis.

# INTRODUCTION

Psoriasis is a prevalent chronic inflammatory skin disease and has a high relapse rate. Its incidence is approximately 0.4% in China and 2–3% in Europe and America (*Rachakonda, Schupp & Armstrong, 2014*). Psoriasis frequently occurs in young adults and always causes hyperkeratosis of epidermal tissue, disappearance of the granular layer, and epidermal abscess. It is accompanied by serious complications such as arthritis, lymph node inflammation, immunity dysfunction, and cardiovascular diseases, leading to a substantial burden for individuals and society (*Griffiths et al., 2021*). Furthermore, psoriasis has a complex pathogenesis, and further research is needed to fully uncover its potential mechanisms.

Recently, macrophage polarization was shown to play a crucial role in the development of inflammation (*Atri, Guerfali & Laouini, 2018*; *Cutolo et al., 2022*). Macrophages can differentiate into two phenotypes: M1 and M2.

Under the influence of various factors, macrophages differentiate into the M1 phenotype, secreting pro-inflammatory factors such as interleukin (IL)-1β, inducible nitric oxide synthase (iNOS), and tumor necrosis factor-α (TNF-α), which contribute to the initiation and aggravation of the inflammatory response. Conversely, M2 macrophages mainly release anti-inflammatory factors such as IL-10, transforming growth factor-β (TGF-β), and arginase 1 (Arg1), inhibiting the inflammatory response (*Shapouri-Moghaddam et al., 2018*; *Wang et al., 2021*).

Emerging evidence revealed that polarization of macrophages towards the M1 phenotype contributes to psoriasis by activating inflammatory responses and releasing a multitude of pro-inflammatory factors. *Lu et al. (2018)* reported that blocking M1 macrophage polarization attenuates psoriatic inflammation by regulating Toll-like receptors 7, 8, and 9 (TLRs 7-9), suggesting a potential treatment strategy for psoriasis. *Zhang et al. (2016)* reported that IL-35 can inhibit the inflammatory process in psoriasis by decreasing the M1/M2 macrophage ratio. *Zeng et al. (2023)* demonstrated that polarization of macrophages activates resident immune cells in psoriatic tissues by secreting proinflammatory cytokines and contributes to the exacerbation of psoriasis.

In summary, the polarization state of macrophages plays an important role in the pathogenesis of psoriasis. Modulating the differentiation of macrophages towards the M1 phenotype is considered beneficial for the treatment of psoriasis, as it can help reduce inflammation and mitigate the symptoms associated with the condition.

Exosomes, small extracellular vesicles involved in intercellular communication, play a role in carrying various molecules, including microRNAs (miRNAs) (*Yu, Odenthal & Fries, 2016*). Psoriatic keratinocyte-derived exosomes have been identified as key regulators of the immune system in patients with psoriasis. They activate neutrophils and exacerbate skin inflammation (*Jiang et al., 2019*). Furthermore, luteolin attenuated the lesions and symptoms of psoriasis in HaCaT cells by suppressing the expression of HSP90, inhibiting exosome secretion, and regulating the proportion of immunocytes (*Lv et al., 2020*). Exosomes derived from wound-edge keratinocytes were significantly enhanced after injury and were selectively engulfed by wound macrophages (*Zhou et al., 2020a*). Therefore, the crosstalk between immunocytes and psoriatic keratinocytes-derived exosomes plays a crucial role in the progression of psoriasis.

The vitamin D receptor (VDR), a nucleophilic protein belonging to the nuclear receptor superfamily, mediates the biological effects of the active form of vitamin D3, 1,25-dihydroxy vitamin D3 [$1,25(OH)_2D3$] (*Ayala-Fontánez, Soler & McCormick, 2016*; *Hu, Bikle & Oda, 2014*; *Woo, Cho & Park, 2017*). In patients with psoriasis, VDR is expressed abnormally low in keratinocytes, and its expression levels are significantly negatively correlated with the severity of the disease (*Chandra, Roesyanto-Mahadi & Yosi, 2020*). Furthermore, the activation of VDR inhibits psoriasis-like skin inflammation by suppressing signal transducer and activator of transcription (STAT) signaling pathway (*Gao et al., 2020b*). Additionally, VDR inhibits the proliferation and promotes the differentiation of macrophages (*O'Kelly et al., 2002*).

However, it is still unclear whether VDR-deficient keratinocytes-derived exosomes (Exos-shVDR) affect the biological functions of macrophages. We aimed to provide an answer to this question and offer a foundation for exploring new diagnostic and therapeutic targets for psoriasis.

Herein, we discovered that Exos-shVDR promoted macrophage proliferation and their polarization towards the M1 phenotype, and inhibited their apoptosis, thereby promoting the development of psoriasis. Mechanistically, these effects of Exos-shVDR were mediated through miR-4505.

# MATERIALS AND METHODS

## Cell culture

The HaCaT (iCell-h066) and THP-1 human monocytic leukemia (THP-1 cells, iCell-h213) cell lines were purchased from iCell Bioscience Inc. (Shanghai, China). The cells were cultured in Dulbecco's modified Eagle medium (DMEM) (PM150210; Procell, Wuhan, China) supplemented with 10% fetal bovine serum (42G3279K; Gibco, Beijing, China) and 100 U/mL penicillin/streptomycin (15140122; Gibco, Beijing, China) in an incubator at 37 °C with 5% $CO_2$. THP-1 cells were differentiated into macrophages by treating them with 100 ng/mL phorbol 12-myristate 13-acetate (HY-18739; PMA, MedChemExpress, Monmouth Junction, NJ, USA) for 48 h prior to exosome uptake.

## Transfection

The short hairpin RNA (shRNA) targeting VDR (shVDR#1, shVDR#2, and shVDR#3), miR-4505 mimics, miR-4505 inhibitor, and their negative controls (shNC, NC mimics, and NC inhibitor) were obtained from RiboBio Corporation (Guangzhou, China). Transient transfection was performed using Lipofectamine 8000™ (C0533; Beyotime, Shanghai, China) following the manufacturer's instructions. VDR shRNAs were transfected into keratinocytes using Lipofectamine 8000™ (C0533; Beyotime, Shanghai, China) for 24 h to achieve VDR deficiency, as confirmed by reverse transcription quantitative polymerase chain reaction (RT-qPCR). Additionally, the miR-4505 inhibitor was transfected into exosomes using Exo-Fect™ Exosome Transfection Kit (EXFT10A-1; System Biosciences, Palo Alto, CA, USA). The transfection efficiency was verified by RT-qPCR. The utilized sequences of shRNA, mimics, and inhibitors are provided below:

shVDR#1: CCGGTCGAAGTGTTTGGCAATGAGATCTCGAGATCTCATTGCCAAACACTTCGTTTTTGAATT;

shVDR#2: CCGGTCCTCCAGTTCGTGTGAATGATCTCGAGATCATTCACACGAACTGGAGGTTTTTGAATT;

shVDR#3: CCGGTGTCATCATGTTGCGCTCCAATCTCGAGATTGGAGCGCAACATGATGACTTTTTGAATT;

scramble: CCGGTCCTAAGGTTAAGTCGCCCTCGCTCGAGCGAGGGCGACTTAACCTTAGGTTTTTGAATT;

miR-4505 mimics,
Sense: 5′-AGGCUGGGCUGGGACGGA-3′, Antisense: 5′-CGUCCCAGCCCAGCCUUU -3′;

NC mimics,
Sense: 5′- UUGUACUACACAAAAGUACUG-3′, Antisense: 5′-GUACUUUUGUGUAGUACAAUU-3′;

miR-4505 inhibitor: TCCGUCCCAGCCCAGCCT;
NC inhibitor: CAGUACUUUUGUGUAGUACAA.

## Extraction and characterization of exosomes from HaCaT cells

The exosomes were isolated from HaCaT cells using Total Exosome Isolation Reagent (4478359; Invitrogen, Carlsbad, CA, USA) following the manufacturer's instructions. Briefly, the culture supernatants of HaCaT cells stably transfected with sh-VDR or sh-NC were collected and centrifuged at 2,000 g for 20 min to remove cells and debris. The resulting supernatants were filtered through a 0.22 μm filtering membrane to remove residual impurities and collected in an ultrafiltration cube. An equal volume of Total Exosome Isolation Reagent (4478359; Invitrogen, Carlsbad, CA, USA) was added to the ultrafiltration cube. Finally, the mixture in the ultrafiltration cube was centrifuged at 10,000 g for 1 h, and the exosomes were collected from the sediment. All operating procedures were carried out at 4 °C. The concentration of exosomes was determined using the Pierce™ bicinchoninic acid (BCA) protein assay (23225; Thermo Scientific, Waltham, MA, USA). The characterization of exosomes was assessed through transmission electron microscopy (TEM), nanoparticle tracking analysis (NTA), and assessment of the protein

expression of exosomal markers (CD63, TSG101). For TEM, the exosomes were mixed with 10 μL uranyl acetate for 1 min and photographed using a transmission electron microscope (Hitachi, Chiyoda City, Japan) after a brief drying period. NTA was performed using a particle size analyzer (NanoFCM, Beijing, China) to track the particle size distribution and concentration of the exosomes. Additionally, the protein expression of exosomal markers was measured by Western blot.

## Exosomes uptake assay

The indicated exosomes were labeled using the PKH26 Green Fluorescent Cell Linker Kit (NA.32; Sigma, St. Louis, MI, USA). Briefly, the exosomes were cultured with diluent C and PKH26 dye for 5 min, and then the unincorporated dye was removed using exosome spin columns. The macrophages were incubated in a medium containing PKH26-labeled exosomes. The treated macrophages were stained with 4′,6-diamidino-2-phenylindole (DAPI) and photographed using a fluorescence microscope (Mshot, MF52).

## RT-qPCR

Total RNA was extracted using Trizol reagent (15596-026; Invitrogen, Waltham, MA, USA) and reverse transcribed into cDNA using PrimeScript™ RT reagent kit and gDNA Eraser (RR047A; Takara, Dalian, China). RT-qPCR amplification was performed on an ABI7500 quantitative PCR instrument using the SYBR Green qPCR Mix (D7260; Beyotime, Shanghai, China). U6 and glyceraldehyde-3-phosphate dehydrogenase (GAPDH) were used as housekeeping genes for miRNAs and other mRNAs, respectively. Relative expression levels of miRNAs and mRNAs were calculated by the $2^{-\Delta\Delta Ct}$ method, and all of the RT-qPCR reactions were performed in triplicate. The primers used in the present study were the following:

VDR:
F: 5′- GTGAGCTGAGATCGTGCCGTTA -3′,
R: 5′- GGTCCTGTCCTGGTCCACTTCT -3′;
GAPDH,
F: 5′- AAGTATGACAACAGCCTCAAG -3′,
R: 5′- TCCACGATACCAAAGTTGTC -3′;
miR-4505:
F: 5′- TTATCTTTAGGCTGGGCTGG -3′,
R: 5′-GTCGTATCCAGTGCGTGTC -3′,
RT: 5′- GTCGTATCCAGTGCGTGTCGTGGAGTCGGCAATTGCACTGGAT ACGACGACGGA -3′;
miR-4507:
F: 5′- AACTAAACTGGGTTGGGCTGG -3′,
R: 5′-GTCGTATCCAGTGCGTGTC -3′,
RT: 5′- GTCGTATCCAGTGCGTGTCGTGGAGTCGGCAATTGCACTGGATA CGACGCTGGG -3′;
miR-4563:
F: 5′- CGTGGAGTTAAGGGTTGCT -3′,

R: 5′-GTCGTATCCAGTGCGTGTC-3′,

RT: 5′- GTCGTATCCAGTGCGTGTCGTGGAGTCGGCAATTGCACTGGAT
ACGACTGGAGA -3′.

U6:

F: 5′- CTCGCTTCGGCAGCACATATACT -3′,

R: 5′- ACGCTTCACGAATTTGCGTGTC-3′,

RT: 5′- AAAATATGGAACGCTTCACGAATTTG -3′.

## Western blot

Total proteins from cells or exosomes were isolated using RIPA lysis buffer (R0020; Solarbio, Beijing, China) and quantified using a BCA detection kit (P0009; Beyotime, Beijing, China) according to the manufacturer's instructions. Proteins with suitable concentrations were separated using sodium dodecyl-sulfate polyacrylamide gel electrophoresis and transferred to polyvinylidene fluoride membranes (WGPVDF22; Servicebio, Wuhai, China). The membranes were incubated overnight at 4 °C with the following primary antibodies: anti-CD63 (ab134045; Abcam, Shanghai, China), anti-TSG101 (ab30871; Abcam, Shanghai, China), anti-VDR (PA5-109276; Invitrogen, Waltham, MA, USA), and anti-GAPDH (GB11002; Servicebio, Wuhai, China), after blocking with 5% blocking solution. Subsequently, the membranes were incubated with a secondary antibody (IH-0011; DingGuo, Beijing, China) at room temperature for 2 h. Protein bands and gray values were visualized and quantified using a gel imaging system (GE Healthcare, Chicago, IL, USA) and Image J software (NIH, Bethesda, MD, USA), respectively.

## Cell counting kit-8 (CCK-8) assay

The viability of THP-1 cells was assessed using the CCK-8 kit (C0037; Beyotime, Beijing, China) following the manufacturer's instructions. Briefly, the treated macrophages were inoculated in a 96-well plate. CCK-8 solution was added to each well and incubated in the dark for 2 h. The absorbance values at 450 nm were measured using a spectrophotometer (Tecan, Männedorf, Switzerland).

## Flow cytometry

Flow cytometry was used to evaluate cell apoptosis and macrophage polarization. For cell apoptosis analysis, the treated cells were incubated with annexin V conjugated with fluorescein isothiocyanate (FITC) and propidium iodide (PI) (C1062L; Beyotime, Beijing, China) at room temperature in the dark for 20 min. For the macrophage polarization, the treated cells were fixed with 1% paraformaldehyde (PFA) overnight at 4 °C and then incubated with anti-CD86 (ab239075; Abcam, Shanghai, China) and anti-CD206 (ab300621; Abcam, Shanghai, China) antibodies for 30 min at 20 °C.

Following incubations, the cells were analyzed by flow cytometry (BD, Franklin Lakes, NJ, USA).

## Data collection and analysis

The miRNAs expression data of patients with or without psoriasis were obtained from the Gene Expression Omnibus (GEO) database (accession no. GSE115293). The dataset included four psoriasis skin samples and four normal skin samples. Differential expression analysis of miRNAs between patients with psoriasis and without (controls) was performed using the "edgeR" package with the following filter criteria: |log (fold change, FC)| > 1.5 and adj. $P$ levels < 0.05. Volcano plots were generated to illustrate the differentially expressed miRNAs.

## Statistical analysis

Statistical analyses and image exports were performed using GraphPad Prism 9.0 (GraphPad Software, San Diego, CA, USA). The Student's t-test and one-way analysis of variance (ANOVA) were used to compare the results and evaluate the differences between groups. A $p$-value < 0.05 was considered statistically significant.

# RESULTS

## Characterization of Exos-shNC and Exos-shVDR

To silence VDR expression in HaCaT cells, three shRNAs targeting VDR (shVDR#1, shVDR#2, shVDR#3) and negative control (shNC) were constructed and transfected into HaCaT cells. The expression levels of VDR were detected after transfection by RT-qPCR and Western blot. The three shVDRs significantly suppressed the expression of VDR in HaCaT cells. Among them, shVDR#1 (shVDR) demonstrated the most significant inhibitory effect and was selected for the subsequent experiments (Fig. 1A).

Subsequently, HaCaT cells stably expressing shVDR and shNC were constructed by lentiviral transduction of shVDR and shNC, respectively. The resulting exosomes, designated as Exos-shVDR and Exos-shNC, were extracted using Total Exosome Isolation reagent. The exosomes were identified by Western blot, TEM, and NTA.

Western blot analysis confirmed the enhanced expression of exosome-specific markers (CD63 and TSG101) in both Exos-shVDR and Exos-shNC, while nearly no expression was observed in the supernatant of HaCaT cells with stable transfected shVDR or shNC treated with exosome inhibitor GW4869 (Fig. 1B).

The morphological characteristics of exosomes were evaluated by TEM, revealing that both Exos-shVDR and Exos-shNC displayed a typical double membrane structure with a particle size of 100 nm (Fig. 1C). Additionally, NTA demonstrated that the particle size of Exos-shVDR and Exos-shNC ranged from 50 to 150 nm (Fig. 1D).

Furthermore, the expression levels of VDR in Exos-shNC and Exos-shVDR were measured by RT-qPCR and Western blot to confirm VDR silencing in exosomes derived from HaCaT cells with stable shVDR transfection. The expression of VDR was significantly lower in Exos-shVDR than in Exos-shNC (Fig. 1E). Taken together, these results demonstrate the successful extraction of exosomes with VDR knocked down, along with their negative control, meeting the requirements for subsequent experiments.
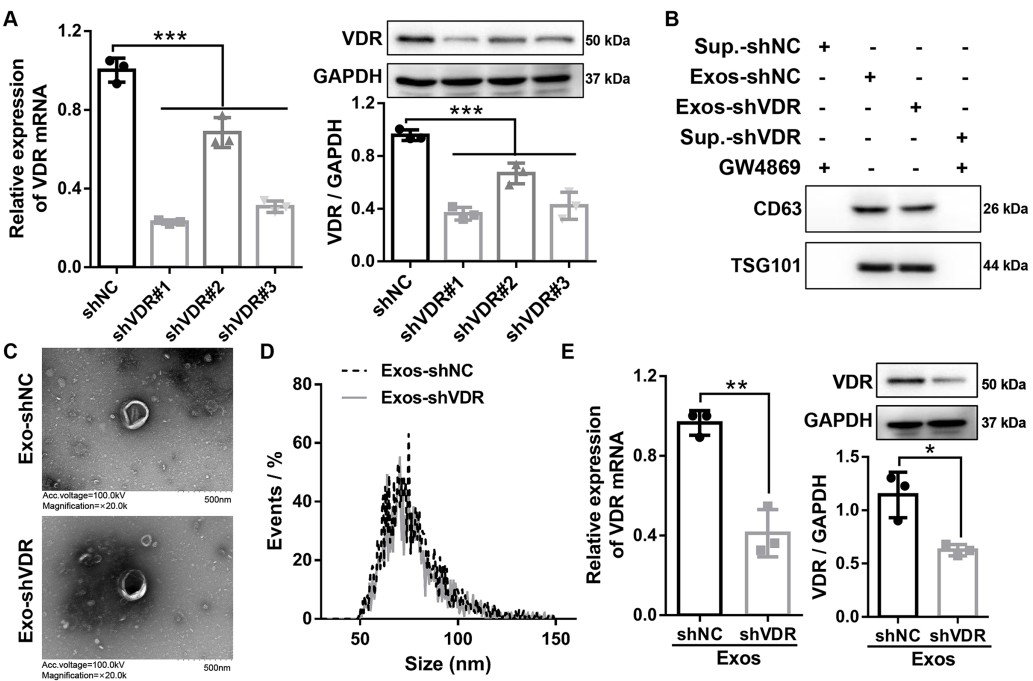

**Figure 1 Characterization of exosomes derived from HaCaT cells with stable transfected shVDR and shNC.** (A and B) The expression levels of VDR were detected by RT-qPCR and Western blot in HaCaT cells transfected with shVDR #1, shVDR #2, shVDR #3, and shNC. $N = 3$. (B) Western blot was applied to evaluate the protein expression levels of exosome-specific markers (CD63, TSG101) in exosome and supernatants. $N = 3$. (C) The morphological characteristics of exosomes were evaluated by transmission electron microscopy. $N = 2$. (D) The particle size of exosomes was evaluated by transmission electron microscopy and nanoparticle tracking analysis (NTA). $N = 1$. (E) The expression of VDR in exosomes derived from HaCaT cells with or without VDR knockdown was measured by RT-qPCR and Western blot. $N = 3$. $^*P < 0.05$, $^{**}P < 0.01$, and $^{***}P < 0.001$. NC, negative control; shRNA, short hairpin RNA; VDR, vitamin D receptor; RT-qPCR, real-time quantitative polymerase chain reaction; shNC, shRNA of negative control; shVDR, the short hairpin RNA target of VDR; Exos-shNC, exosomes derived from HaCaT cells with stably transfected shNC; Exos-shVDR, exosomes derived from HaCaT cells with stably transfected shVDR.

## Exos-shVDR promoted the proliferation of macrophages, their polarization towards the M1 phenotype, and inhibited their apoptosis

To explore the function of Exos-shVDR in macrophages, THP-1 cells were differentiated into macrophages using 100 ng/mL PMA for 48 h. Subsequently, they were co-cultured with PKH26-labeled Exos-shVDR or PKH26-labeled Exos-shNC. The results demonstrated the uptake of both Exos-shVDR and Exos-shNC by macrophages after 24 h of co-culture (Fig. 2A).

Following treatment of PMA-induced macrophages with Exos-shVDR or Exos-shNC for 24 h, cell proliferation and apoptosis were assessed using CCK-8, Western blot, and flow cytometry, as well as by assessing the activity of caspase-3. The CCK-8 assay revealed that Exos-shVDR significantly enhanced macrophage proliferation compared to Exos-shNC (Fig. 2B). Flow cytometry using AV-FITC/PI staining demonstrated Exos-shVDR significantly inhibited macrophage apoptosis compared to Exos-shNC (Fig. 2C).

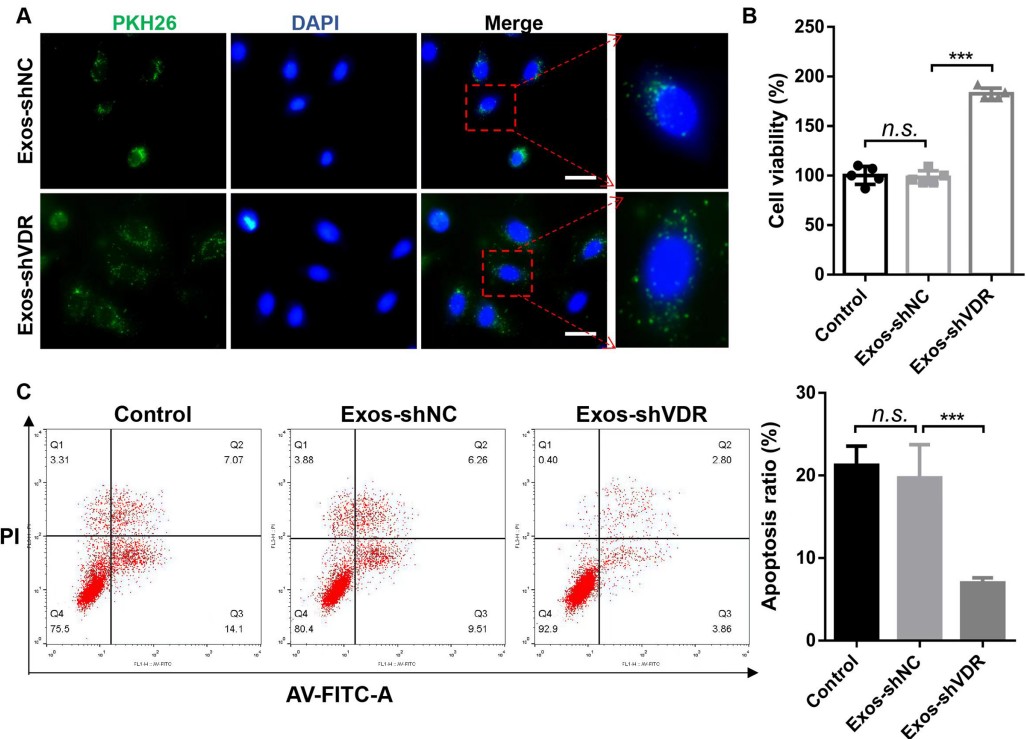

**Figure 2 Exosomes derived from HaCaT cells with stable transfected shVDR (Exos-shVDR) display pro-proliferative and anti-apoptotic effects in macrophage.** (A) The degree of uptake of exosomes by macrophages was tracked by PKH26-labled exosomes staining. Magnification ×40. Scale bar = 200 μM. $N = 3$. (B) Proliferation ability was evaluated by the CCK-8 assay. $N = 5$. (C) Apoptosis ratio was detected by flow cytometry with AV-FITC/PI staining. $N = 3$. [n.s.]$P > 0.05$, [***]$P < 0.001$. Exos-shNC, exosomes derived from HaCaT cells with stable transfected shNC; Exos-shVDR, exosomes derived from HaCaT cells with stable transfected shVDR.               

To evaluate the effect of Exos-shVDR on macrophage M1/M2 polarization, we used flow cytometry, RT-qPCR, and ELISA to quantify the expression levels of the markers of macrophage polarization in macrophages treated with Exos-shVDR or Exos-shNC for 24 h. Exos-shVDR significantly promoted the expression levels of the M1 polarization marker CD86, while it had no significant effect on the expression levels of the M2 polarization marker CD206 (Fig. 3A).

Consistent with the result of flow cytometry, RT-PCR analysis revealed that Exos-shVDR significantly increased the mRNA expression levels of M1 polarization markers (IL-1β, TNFα, and IL-6), while it had no significant effect on the mRNA expression of the M2 polarization marker IL-10 (Fig. 3B). Furthermore, ELISA demonstrated that Exos-shVDR significantly increased the levels of expression of M1 polarization markers (IL-1β, TNFα, and IL-6) in the supernatant of macrophages compared to control. However, it had no significant influence on the level of expression of macrophage M2 polarization marker IL-10 (Fig. 3C). Exos-shNC had no impact on macrophage proliferation, apoptosis, and M1/M2 polarization (Figs. 2 and 3). Overall, our results demonstrated that Exos-shVDR promotes macrophage proliferation, their polarization towards the M1 phenotype, and inhibits their apoptosis.

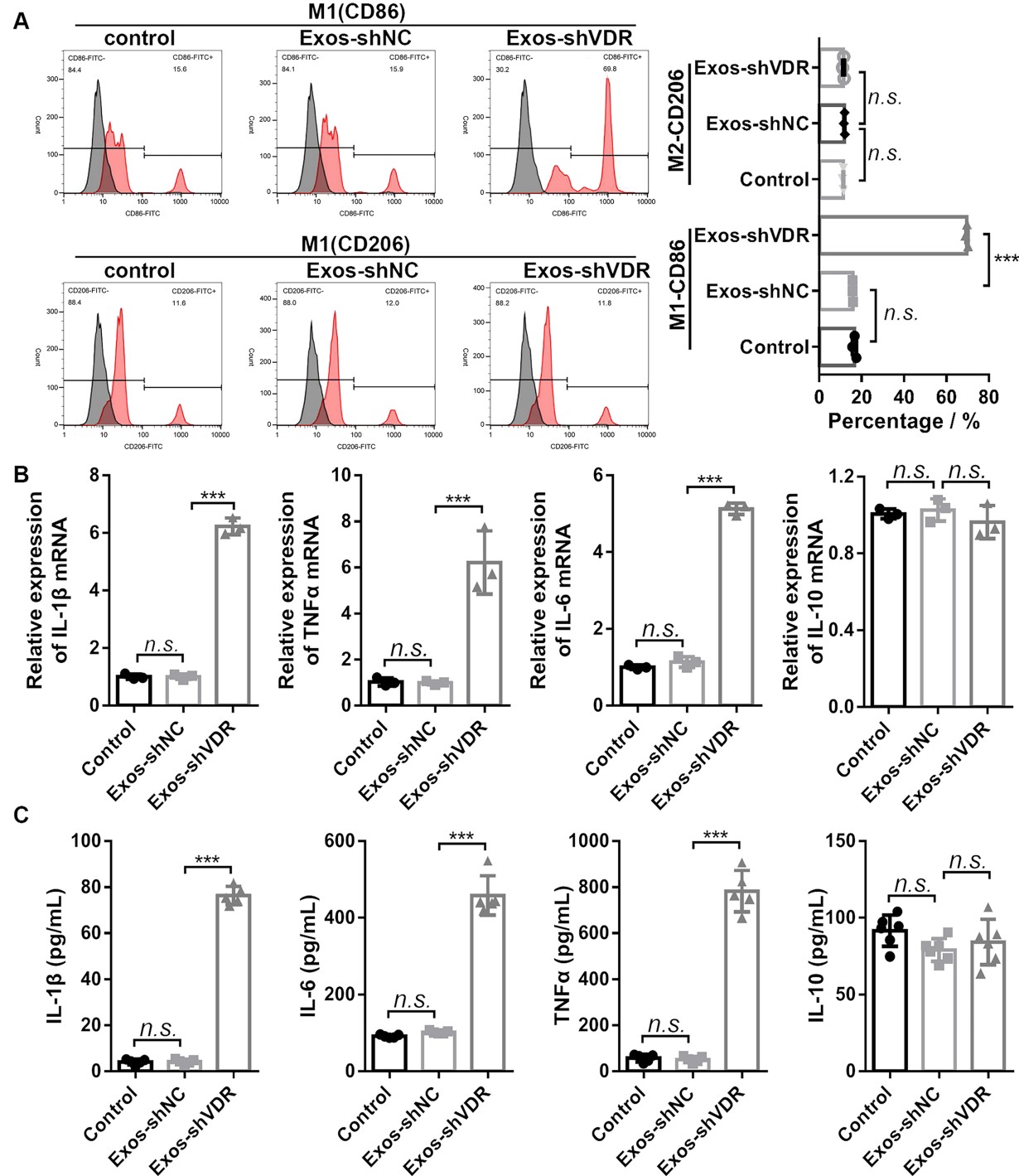

**Figure 3 Exos-shVDR promoted macrophage M1 polarization.** (A) The expression levels of macrophage M1 polarization marker (CD86) and M2 polarization marker (CD206) in macrophages were measured by flow cytometry. $N = 3$. (B) The mRNA expression levels of macrophage M1 polarization markers (IL-1β, TNFα, IL-6) and M2 polarization marker (IL-10) in macrophages were detected by RT-qPCR. $N = 3$. (C) ELISA was applied to detect the content of macrophage M1 polarization markers (IL-1β, αTNFα, IL-6) and M2 polarization marker (IL-10) in the supernatants of macrophages. $N = 5$. $^{n.s.}P > 0.05$, $^{***}P < 0.001$. NC, negative control; shRNA, short hairpin RNA; VDR, vitamin D receptor; mRNA, messenger RNA; IL, interleukin; TNF, tumor necrosis factor; RT-qPCR, real-time quantitative polymerase chain reaction; ELISA, enzyme-linked immunosorbent assay; Exos-shNC, exosomes derived from HaCaT cells with stably transfected shNC; Exos-shVDR, exosomes derived from HaCaT cells with stably transfected shVDR.

### High expression of miR-4505 in Exos-shVDR promotes macrophage proliferation and their polarization towards M1 while inhibiting macrophage apoptosis

Evidence indicates that miRNAs carried by exosomes play a role in various diseases including cancer (*Wang et al., 2022*; *Zheng et al., 2020*), stroke (*Chen et al., 2015*; *Zhao et al., 2013*), and diabetes (*Fluitt et al., 2022*; *Venkat et al., 2019*). To determine the underlying molecular mechanism by which Exos-shVDR promotes polarization towards M1 phenotype, we collected the miRNA expression profile (GSE115293) associated with psoriasis from the GEO database and performed differential expression profiling of miRNA.

A volcano plot revealed 26 dysregulated miRNAs in psoriatic skin tissue compared to normal skin tissue, with 12 upregulated and 14 downregulated miRNAs (Fig. 4A). Among the significantly upregulated miRNAs, we selected miR-4507, miR-4653, and miR-4505 for further analysis of their expression in Exos-shVDR and Exos-shNC. RT-qPCR demonstrated that miR-4507 was significantly reduced, while miR-4653 and miR-4505 were significantly increased in Exos-shVDR compared to Exos-shNC (Fig. 4B). Due to the higher difference in miR-4505 expression between Exos-shVDR and Exos-shNC compared to that in miR-4653-3p, the former was selected for further study. Treatment of macrophages with Exos-shVDR and Exos-shNC for 24 h revealed a significantly higher level of expression of miR-4505 in Exos-shVDR-treated macrophages than in Exos-shNC-treated macrophages (Fig. 4C). These results revealed that miR-4505 is highly expressed in Exos-shVDR and may serve as a key molecule in the regulation of macrophage function.

To further explore the function of miR-4505 in macrophages, macrophages were transfected with miR-4505 mimic or NC mimic for 24 h, followed by RT-qPCR, CCK-8 assay, Western blot, caspase-3 activity assessment, and flow cytometry. The level of expression of miR-4505 was significantly increased in macrophages transfected with miR-4505 mimic compared to those transfected with NC mimic (Fig. 4D). CCK-8 assay revealed that the overexpression of miR-4505 significantly enhanced macrophage proliferation (Fig. 4E). Flow cytometry indicated that the overexpression of miR-4505 significantly reduced macrophage apoptosis (Fig. 4F).

To further explore the impact of miR-4505 on macrophage polarization, RT-qPCR and ELISA were used to quantify the levels of expression of macrophage polarization-related markers. RT-PCR revealed that the upregulation of miR-4505 significantly increased the mRNA expression levels of M1 polarization markers (IL-1$\beta$, TNF$\alpha$, and IL-6), while did not affect the mRNA expression of M2 polarization marker IL-10 (Fig. 4G). Similarly, the results of ELISA indicated that increased levels of miR-4505 significantly increased the secretion of M1 polarization markers (IL-1$\beta$, TNF$\alpha$, and IL-6) in the treated-macrophage supernatant, while exhibiting no significant impact on the level of M2 polarization marker IL-10 (Fig. 4H). Overall, our results indicate that the overexpression of miR-4505 promotes macrophage proliferation and their polarization towards the M1 phenotype, while inhibiting macrophage apoptosis.

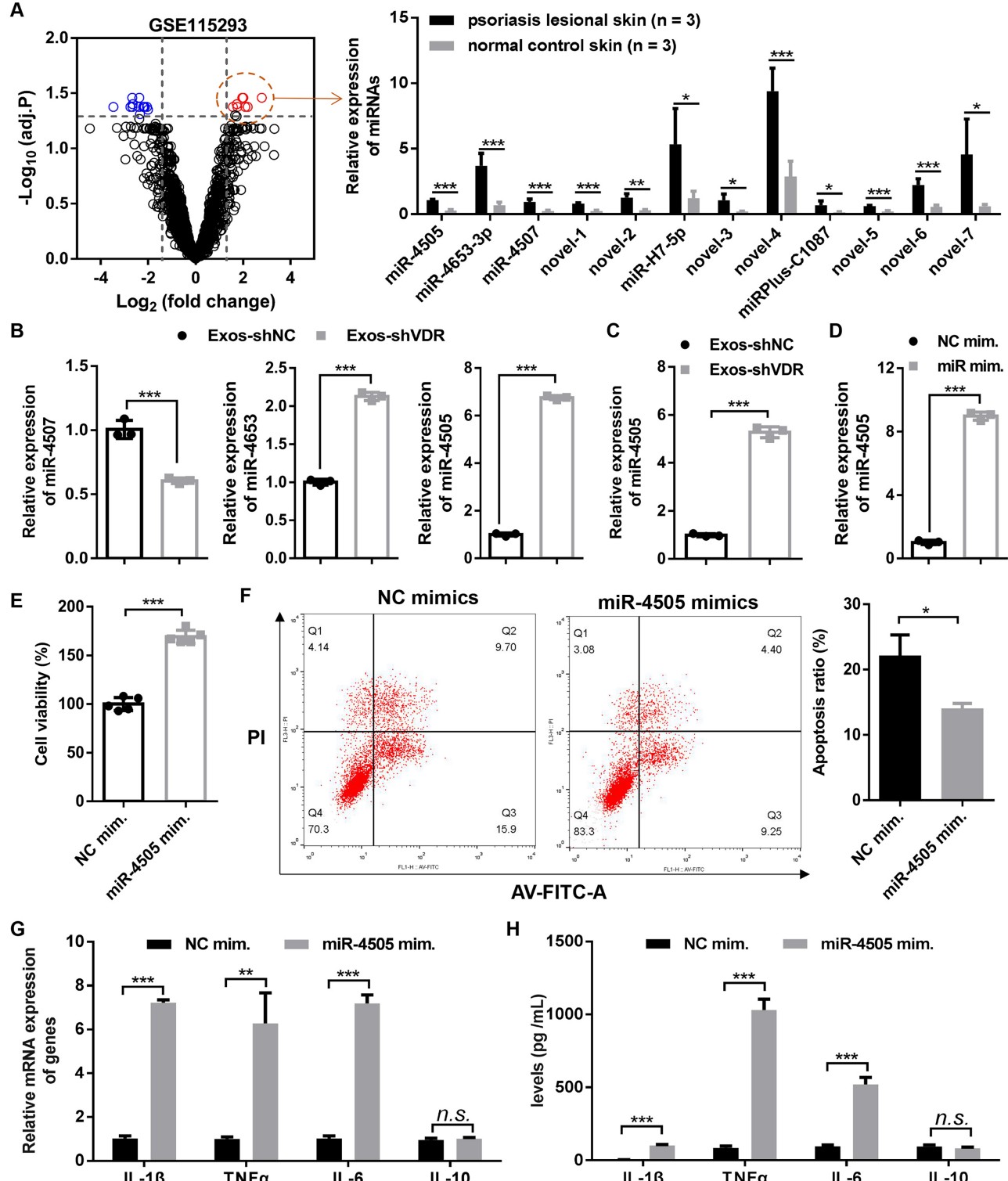

**Figure 4 The high expression of miR-4505 in Exos-shVDR promoted macrophage proliferation and M1 polarization while inhibiting macrophage apoptosis.** (A) A volcano plot showing the miRNA expression profile between the psoriatic skin tissues and the normal skin tissues in the GSE115293 expression matrix and bar graphs showing the expression levels of 12 abnormally upregulated miRNAs in normal skin tissue and psoriatic skin tissue. Red circles represent significantly up-regulated miRNAs, blue circles represent significantly down-regulated miRNAs, and black circles represent no differentially expressed miRNAs. *N* = 3. (B) RT-qPCR was performed to evaluated the differential expression of miR-4507,

**Figure 4 (continued)**
miR-4653, and miR-4505 between Exos-shVDR and Exos-shNC. N = 3. (C) The expression of miR-4505 in macrophages treated with Exos-shVDR or Exos-shNC was detected by RT-qPCR. N = 3. (D) The expression of miR-4505 in macrophages transfected with miR-4505 mimics or NC mimics was assessed by RT-qPCR. N = 3. (E) Macrophage proliferation ability was detected by the CCK-8 assay. N = 5. (F) Apoptosis was assessed by flow cytometry with AV-FITC/PI staining. N = 3. (G) The expression levels of M1 macrophage markers (IL-1β, TNFα, and IL-6) and M2 macrophage markers (IL-10) in macrophage were assessed by RT-qPCR. N = 3. (H) ELISA was performed to evaluate the content of M1 macrophage markers (IL-1β, TNFα, and IL-6) and M2 macrophage markers (IL-10) in the supernatant of macrophages. N = 5. $^{n.s.}P > 0.05$, $^*P < 0.05$, $^{**}P < 0.01$, and $^{***}P < 0.001$. NC, negative control; shRNA, short hairpin RNA; VDR, vitamin D receptor; mRNA, messenger RNA; miRNA, microRNA; IL, interleukin; TNF, tumor necrosis factor; RT-qPCR, real-time quantitative polymerase chain reaction; ELISA, enzyme-linked immunosorbent assay; AV-FITC/PI, Annexin V-fluorescein isothiocyante/propidium iodide; Exos-shNC, exosomes derived from HaCaT cells with stably transfected shNC; Exos-shVDR, exosomes derived from HaCaT cells with stably transfected shVDR; NC mim., negative control mimics; miR-4505 mim., miR-4505 mimics.                                                                      

## Exos-shVDR effect on macrophage is mediated through miR-4505

To determine whether miR-4505 is involved in the regulation of macrophage function by Exos-shVDR, Exos-shVDR was transfected with either NC inhibitor or miR-4505 inhibitor using the Exo-Fect™ Exosome Transfection kit. The results demonstrated that the expression level of miR-4505 in Exos-shVDR transfected with miR-4505 inhibitor (Exos-shVDR-miR-4505 inhibitor) was significantly lower than that in Exos-shVDR transfected with NC inhibitor (Exos-shVDR-NC inhibitor), indicating effective inhibition of miR-4505 in Exos-shVDR transfected with miR-4505 inhibitor (Fig. 5A).

To investigate the role of miR-4505 in mediating the effects of Exos-shVDR on macrophages, we treated macrophages with Exos-shVDR-miR-4505 inhibitor and Exos-shVDR-NC inhibitor for 24 h. RT-qPCR was used to quantify the expression level of miR-4505 in macrophages. The level of expression of miR-4505 in macrophages treated with Exos-shVDR-miR-4505 inhibitor was significantly lower than in macrophages treated with Exos-shVDR-NC inhibitor (Fig. 5B). CCK-8 assay indicated that Exos-shVDR-miR-4505 inhibitor significantly reduced macrophage proliferation compared to Exos-shVDR-NC inhibitor (Fig. 5C).

Flow cytometry revealed that the Exos-shVDR-miR-4505 inhibitor significantly increased macrophage apoptosis compared to Exos-shVDR-NC inhibitor (Fig. 5D). Moreover, RT-qPCR revealed that the Exos-shVDR-miR-4505 inhibitor dramatically decreased the mRNA expression levels of M1 polarization markers (IL-1β, TNFα, and IL-6) in macrophage compared to Exos-shVDR-NC inhibitor (Fig. 5E). Similarly, ELISA demonstrated that Exos-shVDR-miR-4505 inhibitor dramatically decreased the level of expression of M1 polarization markers (IL-1β, TNFα, and IL-6) in the supernatant of treated macrophages compared to Exos-shVDR-NC inhibitor (Fig. 5F). These results collectively demonstrate that Exos-shVDR promotes macrophage proliferation and polarization towards M1 phenotype, while inhibiting macrophage apoptosis via miR-4505.

## DISCUSSION

Psoriasis is a common skin disease with a hereditary component. It is characterized by abnormal growth and differentiation of keratinocytes, expansion of microvessels in the superficial dermis, and infiltration of inflammatory cells (*Lowes, Bowcock & Krueger,*

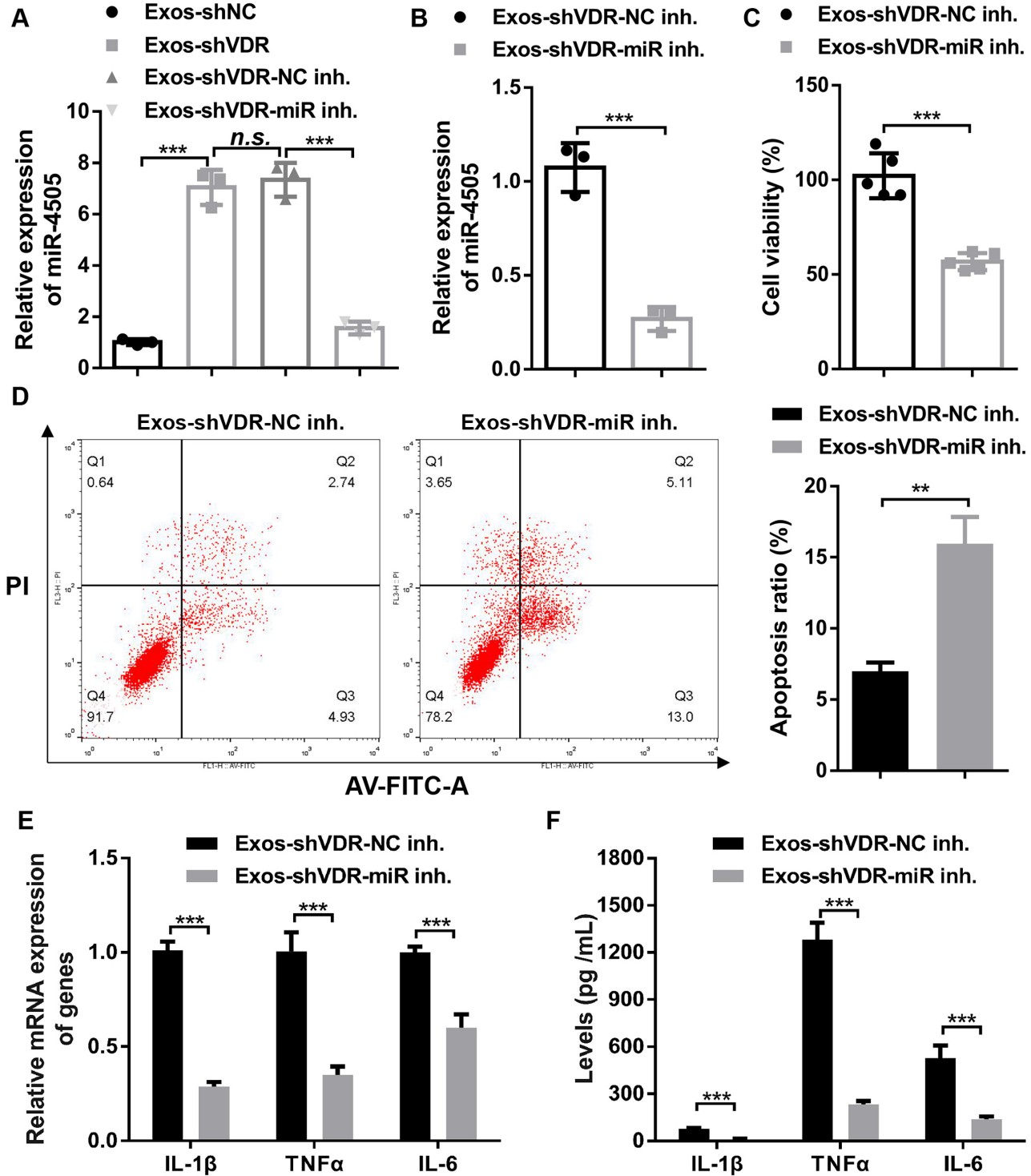

**Figure 5 Exos-shVDR promoted macrophage proliferation and M1 polarization while inhibiting macrophage apoptosis via transferring miR-4505.** (A) Exos-shVDR was transfected with NC inhibitor and miR-4505 inhibitor using the Exo-Fect Exosome Transfection kit, and RT-qPCR was applied to assess the expression of miR-4505. $N = 3$. (B) Exos-shVDR-miR-4505 inhibitor and Exos-shVDR-NC inhibitor were applied to treat macrophages, and the expression of miR-4505 in macrophages was detected by RT-qPCR. $N = 3$. (C) Macrophage proliferation was measured by the CCK-8 assay. $N = 5$. (D) Macrophages apoptosis was assessed by flow cytometry with AV-FITC/PI staining. $N = 3$. (E) The expression levels of M1
**Figure 5 (continued)**
macrophage markers (IL-1β, TNFα, and IL-6) in macrophages were evaluated by RT-qPCR. $N = 5$. (F) ELISA was applied to evaluate the content of M1 macrophage markers (IL-1β, TNFα, and IL-6) in the supernatant of macrophages. $N = 3$. [n.s.]$P > 0.05$, [**]$P < 0.01$, and [***]$P < 0.001$. NC, negative control; shRNA, short hairpin RNA; VDR, vitamin D receptor; mRNA, messenger RNA; miRNA, microRNA; IL, interleukin; TNF, tumor necrosis factor; RT-qPCR, real-time quantitative polymerase chain reaction; ELISA, enzyme-linked immunosorbent assay; AV-FITC/PI, Annexin V-fluorescein isothiocyante/propidium iodide; Exos-shVDR-NC inhibitor, Exos-shVDR transfected with NC inhibitor; Exos-shVDR-miR-4505 inhibitor, Exos-shVDR transfected with miR-4505 inhibitor.               

*2007*). While the exact cause of psoriasis is not fully elucidated, it is believed that abnormal immune function, especially the excessive secretion of inflammatory factors such as IL-1β and IL-6, contributes substantially to its development (*Aleem & Tohid, 2018*).

One prominent feature of psoriasis is the infiltration of macrophages at the dermal-epidermal junction and the release of inflammatory factors. Macrophages play a crucial role in the progression of psoriasis (*van den Oord & de Wolf-Peeters, 1994*; *Vestergaard et al., 2004*) and are a major source of TNF-α in psoriatic lesions. Consequently, anti-TNF-α agents are effective in treating psoriasis (*Kircik & Del Rosso, 2009*). Additionally, inhibiting macrophage polarization into the M1 phenotype mitigates the TLRs-mediated psoriatic inflammation (*Lu et al., 2018*). PSORI-CM02 formula reduced imiquimod-induced macrophage infiltration and M1 polarization in mice, indicating it may possess therapeutic action in psoriasis treatment (*Li et al., 2020*).

Keratinocytes, as the main components of the epidermis, play a crucial role in the pathogenesis of psoriasis by maintaining a mechanical barrier, participating in the initiation and maintenance of skin immune responses, and interacting with immune cells. Keratinocyte-derived exosomes from patients with psoriasis activated neutrophils and aggravated skin inflammation (*Jiang et al., 2019*). Luteolin alleviated the symptoms of psoriasis by reducing the level of expression and exosome secretion of HSP90, as well as regulating the proportion of immunocytes (*Lv et al., 2020*). These studies highlight the significant involvement of keratinocyte-derived exosomes in psoriasis progression. However, the underlying mechanism by which keratinocytes-derived exosomes mediate immunocytes involved in psoriasis remains unclear.

Furthermore, vitamin D, which is synthesized, activated, and degraded independently, in the skin, has been implicated in psoriasis. Patients with psoriasis often have insufficient levels of serum 25(OH)$_2$D$_3$ (*Amon et al., 2018*). VDR is indispensable for the normal biological function of vitamin D3, and its deficiency can exacerbate inflammation, increase cell proliferation, and promote psoriasis development. The vitamin D3 analog calcipotriol attenuates inflammation in psoriatic skin by inhibiting the pivotal IL-23/IL-17 axis and neutrophil infiltration through the VDR signaling pathway (*Germán et al., 2019*). Furthermore, VDR can directly impact the functions of macrophages, such as polarization (*Zhang et al., 2015*; *Zhou et al., 2020b*; *Zhu et al., 2019*) and autophagy (*Kumar et al., 2021*). However, it remains unknown whether VDR-knockdown keratinocyte-derived exosomes regulate macrophage polarization. Herein, exosomes released from VDR knockdown HaCaT cells promoted macrophage proliferation and their polarization towards M1 phenotype while inhibiting macrophage apoptosis.

miRNAs are small noncoding RNAs ranging in length from 18 to 24 nucleotides and playing a crucial role in the occurrence and development of various diseases, including psoriasis (*Naveed et al., 2017*). Thus, a decreased level of miR-1910-3p abolishes the suppressive effect of its target gene IL-17A, accelerating the progression of psoriasis by increasing the levels of pro-inflammatory molecules and promoting the proliferation of keratinocytes (*Karabacak et al., 2021*). miR-125a-5p was shown to be downregulated in psoriasis, and its upregulation increased the activation of the TGFβ/Suppressor of Mothers Against Decapentaplegic (STAT) transcription factor pathway and aggravated the development of psoriasis (*Qu, Liu & Wang, 2021*). In HaCaT cells, miR-125a negatively regulated the IL-23R/JAK2/STAT3 pathway, suppressing cell proliferation and promoting apoptosis (*Su, Jin & Liu, 2021*).

Recently, exosomal miRNAs have been identified, and extensively studied for their biological functions upon transfer to recipient cells. In gastric cancer, macrophage-derived exosomal transfer has been associated with doxorubicin resistance (*Gao et al., 2020a*). Furthermore, the delivery of miRNA-29a-3p by exosomes derived from engineered human mesenchymal stem cells enhanced tumor inhibition by reducing cell migration and vasculogenic mimicry in glioma (*Zhang et al., 2021*).

Based on GEO data mining and experimental verification, we demonstrated that miR-4505 was highly expressed in the keratinocytes of patients with psoriasis. Furthermore, we also observed significantly higher levels of expression of miR-4505 in exosomes derived from VDR knockdown HaCaT cells compared to control HaCaT cells. VDR is a ligand-activated transcription factor belonging to the nuclear receptor superfamily. Previous studies have shown that VDR/retinoid X receptor can modulate the expression of miR-22, miR-296-3p, and miR-498 by regulating the maturation of pre-miRNAs upon binding to VDR response element (*Zenata & Vrzal, 2017*). In addition, *Ge et al. (2020)* reported that VDR bound to the promoter of the miR-27a/b gene to promote miR-27a/b expression in oral lichen planus. Herein, we demonstrated that VDR regulates the expression of miR-4505 in keratinocytes at the transcriptional level, although the specific mechanism requires further verification. The effects of miR-4505 were also investigated in other studies. For example, *Zhang et al. (2018)* revealed that miR-4505 aggravated LPS-induced vascular endothelial damage by inhibiting heat shock protein A12B. *Zhang et al. (2017)* demonstrated that the downregulation of miR-4505 by XLOC_000090 enhances the migration and invasion of lung cancer cells. Furthermore, Lcn2-mediated miR-4505 inhibited oral cancer metastasis and migration by targeting carbonic anhydrase IX (*Lin et al., 2016*). However, the regulatory role of miR-4505 on macrophages was not reported up to now.

Our findings revealed that miR-4505 promoted macrophage proliferation and their polarization towards M1 phenotype, while inhibiting macrophage apoptosis. Furthermore, we demonstrated that exosomes derived from VDR knockdown keratinocytes exert their effects through miR-4505. Our results indicate that miR-4505 might be a new potential therapeutic target for psoriasis. However, this study also has certain limitations: (1) the downstream mechanism of miR-4505 regulation in macrophage polarization has not been elucidated; (2) the underlying mechanism of VDR knockdown-induced miR-4505

overexpression has not been explored; (3) the results of this study have yet to be validated in animal models as they were limited to *in vitro* experiments.

## CONCLUSION

Our study demonstrated that Exos-shVDR promoted macrophage proliferation and polarization towards the M1 phenotype while inhibiting macrophage apoptosis. Mechanistically, these effects occurred due to the VDR knockdown-induced increase in the level of expression of miR-4505 in the exosomes secreted by keratinocytes. This study provides a foundation for further exploration of novel diagnostic and therapeutic targets for psoriasis.

### Funding

This work was funded by the Natural Science Foundation of Hubei (No. 2018CFB289), the Science Foundation of Health Commission of Hubei Province (No. WJ2019M074), the Science and Technology Planning Project of JingMen (No. 2020YFZD022); the Medical Scientific Research Foundation of Guangdong Province of China (No. A2022425); the Project of Administration of Traditional Chinese Medicine of Guangdong Province of China (No. 20221280); The Science Foundation of Health Commission of Hubei Province (No. W2023Q021) supported the APC. The funders had no role in study design, data collection and analysis, decision to publish, or preparation of the manuscript.

### Grant Disclosures

The following grant information was disclosed by the authors:
Natural Science Foundation of Hubei: 2018CFB289.
Science Foundation of Health Commission of Hubei Province: WJ2019M074.
Science and Technology Planning Project of JingMen: 2020YFZD022.
Medical Scientific Research Foundation of Guangdong Province of China: A2022425.
Project of Administration of Traditional Chinese Medicine of Guangdong Province of China: 20221280.
The Science Foundation of Health Commission of Hubei Province: W2023Q021.

### Competing Interests

The authors declare that they have no competing interests.

### Author Contributions

- Wen Sun conceived and designed the experiments, performed the experiments, analyzed the data, prepared figures and/or tables, authored or reviewed drafts of the article, and approved the final draft.
- Jianqin Chen conceived and designed the experiments, performed the experiments, analyzed the data, authored or reviewed drafts of the article, and approved the final draft.
- Jingting Li performed the experiments, authored or reviewed drafts of the article, and approved the final draft.

- Xiaoguang She performed the experiments, prepared figures and/or tables, and approved the final draft.
- Hu Ma performed the experiments, analyzed the data, authored or reviewed drafts of the article, and approved the final draft.
- Shali Wang analyzed the data, prepared figures and/or tables, and approved the final draft.
- Jing Liu conceived and designed the experiments, analyzed the data, authored or reviewed drafts of the article, and approved the final draft.
- Yuan Yuan conceived and designed the experiments, analyzed the data, prepared figures and/or tables, authored or reviewed drafts of the article, and approved the final draft.

## Data Availability

The data is available at Figshare: Sun, Wen (2023). raw data.7z. figshare. Journal contribution. https://doi.org/10.6084/m9.figshare.22705504.v1.

## Supplemental Information

Supplemental information for this article can be found online at http://dx.doi.org/10.7717/peerj.15798#supplemental-information.

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
