# Peer review of "Vitamin D receptor-deficient keratinocytes-derived exosomal miR-4505 promotes the macrophage polarization towards the M1 phenotype"

_PeerJ, doi:10.7717/peerj.15798_

## Round 0.1 · original submission · Minor Revisions

The reviewers have made many constructive suggestions. I believe the necessary revisions and clarifications will be needed to improve this research further. Please refer to these comments and reviews to revise your paper.

Reviewer 1 ·

Basic reporting

This manuscript presents novel findings on the role of VDR-deficient keratinocytes-derived exosomes in macrophages, providing valuable insights into their contribution to psoriasis progression. The language expression of the manuscript is authentic, the logic is relatively clear, and the literature list is also reasonable. The overall writing level is commendable. However, there are also some minor issues that need to be revised.

Experimental design

1.There are many genes involved in the occurrence and development of psoriasis. Why did the author choose VDR for research? Please describe the reason.
2.How do polarized M1/M2 macrophage populations affect the development of psoriasis? It is recommended to add relevant contents.
3.In the experiment, a set of miRNAs associated with the promotion of M1 polarization in macrophages by Exos-shVDR was identified through miRNA expression profiling analysis, and miR-4505 was chosen for subsequent research. However, the article did not mention any previous literature support or relevant functional studies on miR-4505. To support the selection and significance of miR-4505 in this study, it is recommended to provide more background information and previous research results regarding the involvement of miR-4505 in the regulation of macrophage function.
4.The study shows that Exos-shVDR impacts macrophage function by transmitting miR-4505. However, there is a lack of detailed research and validation regarding the underlying mechanisms. It is suggested to conduct further experiments, such as signaling pathway analysis, expression profiling of target genes, and functional validation, to unravel the specific regulatory mechanisms of Exos-shVDR and miR-4505 on macrophage function.

Validity of the findings

1. This study mentions the significance of miRNAs in the occurrence and development of diseases and indicates the high expression of miR-4505 in exosomes derived from keratinocytes. This is an interesting finding, but providing further explanation of the functional role and potential mechanisms of miR-4505 would enhance readers' understanding of its effects.
2. The results should be described in a more clear manner, including the specific time point at which the cells were collected for CCK-8 assay, apoptosis detection, and PCR analysis.
3.The figure captions in Figure 1B should be aligned,and the term "DAPA" in Figure 2A should be corrected to "DAPI".
4. How many replicates were performed in each group? Please include this information in the manuscript, either in Methods or Legend to Figures.

Reviewer 2 ·

Basic reporting

This study indicates that Exos-shVDR secreted by VDR-deficient keratinocytes promote macrophage proliferation, anti-apoptosis, and M1 polarization through the delivery of miR-4505. This shed light on the importance of the crosstalk between keratinocytes and macrophages in the pathogenesis of psoriasis. Modulating this pathway may represent a novel therapeutic strategy for psoriasis.The manuscript can basically reach the level of publication, but details still need to be checked.
1.The Introduction section introduces the role of macrophage polarization in psoriasis and cites some relevant studies. However, it would be beneficial to clarify that M1 macrophage polarization is associated with pro-inflammatory responses and the release of inflammatory factors, which contribute to psoriatic inflammation.
2.In the Introduction section discussing the role of psoriatic keratinocyte-derived exosomes, it would be useful to mention that exosomes are small extracellular vesicles involved in intercellular communication and are known to carry various molecules, including microRNAs.
3.Please verify the information on the origin and product number of the reagents in the Materials and Methods section.
4.The author indicated that “We used Exos-shVDR and Exos-shNC to treat macrophages for 24h, and found that the expression level of miR-4050 in Exos-shVDR-treated macrophages was significantly higher than that in Exos-shVDR-treated macrophages (Figure 4C).” Please confirm if the description is correct.
5.The discussion needs to summarize the findings of the research combined with previous research, and identify the highlights of the research. The discussion should increase the research progress of VDR and miRNAs in macrophages, in order to highlighting the innovation of this study. In addition, the limitations of the study should be discussed.
6.Please check the DAPD of Figure 2A

Experimental design

no comment

Validity of the findings

no comment

Additional comments

no comment

Reviewer 3 ·

Basic reporting

The paper titled “Vitamin D receptor-deficient keratinocyte-derived exosomal miR-4505 promoted the macrophage M1 polarization” is interesting. The results indicate that Exos-shVDR aggravates macrophage proliferation, antiapoptosis, and M1 polarization by delivering a high abundance of miR-4505, a finding which may provide a theoretical basis for the intervention of macrophages as a target for the treatment of psoriasis. In summary, the article has a complete structure, easy to understand expression, and reasonable citation of references. However, there are still several minor issues that if addressed would significantly improve the manuscript.

Experimental design

2.1 This study explores the role of Exos-shVDR on proliferation, anti-apoptotsis, and M1 polarization in macrophages and the results showed that Exos-shVDR promoted proliferation, anti-apoptotsis, and M1 polarization in macrophages. The author needs to clarify whether VDR can directly affect the function of macrophages to demonstrate whether VDR function in macrophages depending on the transport of keratinocytes exosomes.
2.2 This study mentions the significance of miRNAs in the occurrence and development of diseases and indicates the high expression of miR-4505 in exosomes derived from keratinocytes. This is an interesting finding, but providing further explanation of the functional role and potential mechanisms of miR-4505 would enhance readers' understanding of its effects.
2.3 In Abstract, a detailed description is required for the methods, such as the underlying mechanism of Exos-shVDR on the function of macrophages was elucidated by data mining, bioinformatics, RT-qPCR and rescue experiments.
2.4 In Introduction, the author needs to clarify why miRNA was chosen as a target for the impact of Exo -VDR for research.

Validity of the findings

3.1 The construction method of VDR-deficient keratinocytes should be described in detail.
3.2 Some of the descriptions in the discussion were inappropriate, and the author needs to make revisions. such as “ At present, the role of miR-4505 in the physiology and pathology of the body is still unexplored.”
3.3 Please verify if the DAPD in Fig. 2A is a DAPI. The font size in the image needs to be unified, and the annotations need to be refined.
3.4 Photomicrographs should have internal scale markers.

Additional comments

Please discuss the limitations of this study in the discussion, as is necessary.
In addition, please ensure that, before resubmission, the manuscript is carefully checked for English language and grammatical errors. It is important that the message being conveyed in the manuscript is as unambiguous as possible.

---

## Round 0.2 · accepted · Accept

The concerns of all reviewers have been adequately addressed. This revised article has been strengthened and improved. I am satisfied with the revisions and responses made by the authors. This revised paper is ready to be accepted.